# Consumption-based greenhouse gas emissions accounting with capital stock change highlights dynamics of fast-developing countries

Zhan-Ming Chen [1,2], Stephanie Ohshita[2,3], Manfred Lenzen [4], Thomas Wiedmann [4,5], Magnus Jiborn[6,7], Bin Chen[8,9], Leo Lester[10], Dabo Guan [11,12], Jing Meng[12,13], Shiyun Xu[14,15], Guoqian Chen[16], Xinye Zheng[1], JinJun Xue[9,17], Ahmed Alsaedi[15], Tasawar Hayat[15,18] & Zhu Liu [11,12]

Traditional consumption-based greenhouse gas emissions accounting attributed the gap between consumption-based and production-based emissions to international trade. Yet few attempts have analyzed the temporal deviation between current emissions and future consumption, which can be explained through changes in capital stock. Here we develop a dynamic model to incorporate capital stock change in consumption-based accounting. The new model is applied using global data for 1995–2009. Our results show that global emissions embodied in consumption determined by the new model are smaller than those obtained from the traditional model. The emissions embodied in global capital stock increased steadily during the period. However, capital plays very different roles in shaping consumption-based emissions for economies with different development characteristics. As a result, the dynamic model yields similar consumption-based emissions estimation for many developed countries comparing with the traditional model, but it highlights the dynamics of fast-developing countries.

[1] Department of Energy Economics, School of Economics, Renmin University of China, Beijing 100872, China. [2] Energy Analysis and Environmental Impacts Department, Environmental Energy Technologies Division, Lawrence Berkeley National Laboratory, Berkeley, CA 94720, USA. [3] Department of Environmental Science and Management, University of San Francisco, San Francisco, CA 94117-1080, USA. [4] ISA, School of Physics A28, The University of Sydney, Sydney, NSW 2006, Australia. [5] School of Civil and Environmental Engineering, UNSW, Sydney, NSW 2052, Australia. [6] Department of Economic History, Lund University, Box7083S-220 07 Lund, Sweden. [7] Department of Philosophy, Lund University, Kungshuset Lundagård, S-222 22 Lund, Sweden. [8] State Key Joint Laboratory of Environmental Simulation and Pollution Control, School of Environment, Beijing Normal University, Beijing 100875, China. [9] Economic Research Centre of Graduate School of Economics, Nagoya University, Furo-cho, Chikusa-ku, Nagoya, Japan. [10] The Lantau Group (HK) Limited, Hong Kong SAR, China. [11] Department of Earth System Sciences, Tsinghua University, Beijing 100080, China. [12] Water Security Research Centre, Tyndall Centre for Climate Change Research, School of International Development, University of East Anglia, Norwich NR4 7TJ, UK. [13] Department of Politics and International Studies, University of Cambridge, Cambridge CB3 9DT, UK. [14] China Electric Power Research Institute, Beijing 100192, China. [15] NAAM Research Group, Faculty of Science, King Abdulaziz University, Jeddah 21589, Saudi Arabia. [16] Laboratory of Systems Ecology and Sustainability Science, College of Engineering, Peking University, Beijing 100871, China. [17] Center of Hubei Coordinative Innovation for Emissions Trading System, Wuhan, Hubei, China. [18] Department of Mathematics, Quaid-I-Azam University, Islamabad 44000, Pakistan. Correspondence and requests for materials should be addressed to Z.-M.C. (email: chenzhanming@ruc.edu.cn) or to B.C. (email: chenb@bnu.edu.cn) or to Z.L. (email: zhuliu@tsinghua.edu)

Greenhouse gas (GHG) emissions accounting is an important factor in the design and implementation of climate mitigation policy[1,2]. In addition to production-based GHG emissions accounting (PBA) that has provided a baseline for climate science and policy regimes[3–5], consumption-based accounting (CBA) has attracted increased attention over the past decade[6–10]. CBA studies ground on the opinion that those who benefit from a process should bear (at least some) responsibility for the associated emissions[11]. Accordingly, one of the main tasks of CBA is to estimate the consumption-based emissions of an economy, i.e., the environmental impacts generated during the production and use of its final goods and services[12,13]. Especially, by providing information to incorporate emissions embodied in international trade[14–16], CBA extends the national perspective to a global context and draws attention to the emission transfer problem that troubles global mitigation efforts[17,18].

The international trade inherent in globalization has made multi-regional input-output modeling an appropriate tool to describe global supply chains[19–21] and their associated GHG emissions[22–24]. Within the multi-regional input-output framework, economic outputs are categorized as either intermediate input or final demand (which includes consumption and capital investment). In most traditional CBA studies, global GHG emissions of a given year are allocated to different countries according to their final demands (both consumption and capital investment) and thus current emissions embodied in final demand (usually termed emission footprint) of a country are used to indicate its consumption-based emissions[11,25–29]. By adopting this allocation scheme and the associated indicator, CBA model is significantly simplified, because intertemporal dynamics through the use of past capital to enable current production are ignored and all domestic final demand is attributed to support domestic consumption.

The indicator of emissions embodied in final demand applied in traditional CBA model presents a good reflection of emissions driven by consumption activities for a country if the economy is near a steady state and if its capital is only used to produce domestically consumed goods. Unfortunately, in the real world, most economies are moving away from instead of towards the two conditions. Therefore, a traditional CBA perspective might bring bias to the estimation of consumption-based emissions because the capital dynamics and the international engagement of capital services are overlooked.

As a result, this study revisits the definition of consumption-based emissions and constructs an alternative indicator by acknowledging the role and dynamics of capital stock in the global production structure. A new dynamic model is proposed by endogenizing capital as a dynamic stock within the multi-regional input-output framework, in which the contributions of capital input from different times as well as from different regions are taking into account to measure the emissions generated by consumption activities. The new model is applied to the global economy based on the World Input-Output Database[30] to examine the role of capital in consumption-based GHG emissions accounting. Our results confirm the evolution of global capital stock and the international engagement of capital use have significant impact to the consumption-based emissions, especially for fast-developing countries. To provide clarity for the discussion that follows, the term traditional footprint indicates the GHG emissions embodied in final demand based on traditional CBA model, where capital stock is treated in a static way and it is assumed to support only domestic consumption. The term dynamic footprint, on the other hand, indicates the emissions embodied in consumption goods calculated by our new dynamic CBA model (Fig. 1).

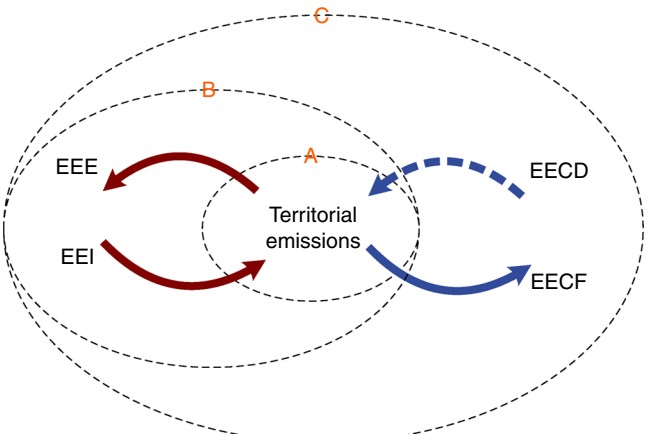

**Fig. 1** Conceptual illustration of embodied emissions flows associated with emissions accounting. EEE represents emissions embodied in export, EEI represents emissions embodied in import, EECF represents emissions embodied in capital formation, and EECD represents emissions embodied in capital depreciation (utilization). Circle A indicates the scope of production-based accounting, which covers emissions within the targeted region during the targeted year. Circle B indicates the scope of traditional CBA, which covers emissions within and without the targeted region during the targeted year. Circle C indicates the scope of dynamic CBA, which covers emissions within and without the targeted region before and during the targeted year. The traditional footprint is calculated as territorial emissions plus EEI less EEE, while the dynamic footprint is calculated as territorial emissions plus EEI and EECD less EEE and EECF. Note that the sizes of the three circles do not necessary represent the relative value of the indicators. For example, territorial emissions of a country can be larger than the traditional footprint when emissions embodied in export exceed those embodied in import

## Results

**Global emissions and footprints**. Given the basic assumption that the emissions intensity of capital stock is not changing dramatically over a short period (which will be shown to be true later), one would expect global PBA is larger than global dynamic CBA footprint when the capital formation exceeds the depreciation of existing capital. This hypothesis is confirmed by the empirical results regardless of the type of GHG or the years studied (Fig. 2). When emissions embodied in international trade are redistributed at the global level, the gap between PBA and dynamic CBA measures indicates the difference between emissions embodied in the formation and depreciation of capital, reflecting a net accumulation of capital in terms of emission embodiment. Corresponding to the influential business cycles that affected the whole world, the gap shrank significantly from 1997 to 2002 and in 2009. On average, the gap between the territorial emissions and the dynamic footprint equaled 7.4% of the former during 1995–2009, and was proportionally larger for $CO_2$ (8.3%) than for $CH_4$ (6.0%) and $N_2O$ (4.1%). This is because capital formation has a closer connection with $CO_2$ emitting activities (e.g., fossil fuel combustion and cement production) than $CH_4$ and $N_2O$ emitting activities (considerable parts of which are agriculture-related).

Under the dynamic CBA analysis, in 2009, the GHG emissions embodied in capital formation is significantly larger than that embodied in depreciation (Fig. 3). This highlights the importance and influence of capital in global production chains. Utilities and heavy industry released almost half of the global territorial GHG emissions in 2009, but only a very small fraction of the world's dynamic footprint came from these two sectors. This is because a majority of their outputs was used as intermediate inputs, not as consumption goods. The emissions emanating from the business

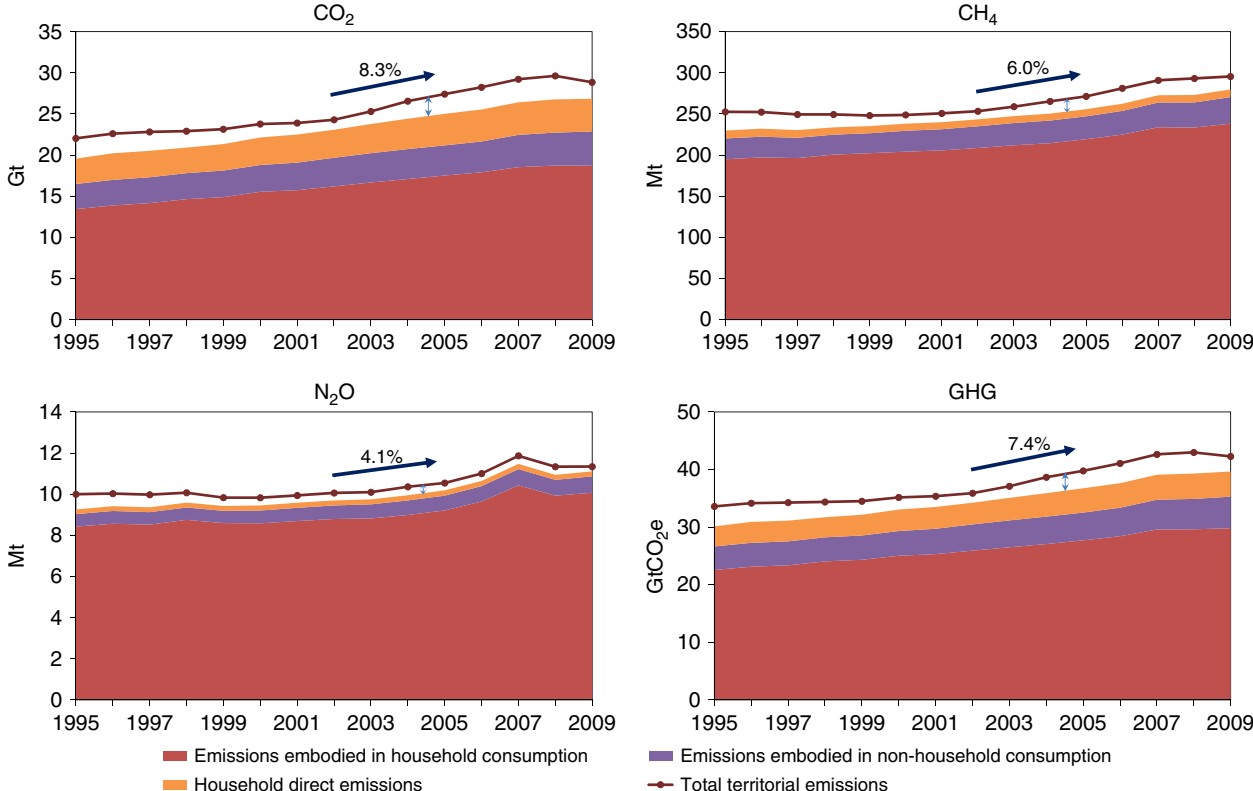

**Fig. 2** Territorial emissions and dynamic footprint of the world. The global dynamic footprint is smaller than the direct territorial emissions during the period and the gap is more remarkable for $CO_2$ than the other two GHG types. The dynamic footprint is the aggregation of household direct emissions, emissions embodied in final household consumption, and emissions embodied in final non-household consumption. GHG emissions are calculated as the sum of $CO_2$, $CH_4$, and $N_2O$ emissions based on the relative global warming potentials of 100 years with climate change feedback[58], which are 1, 34, and 298, respectively

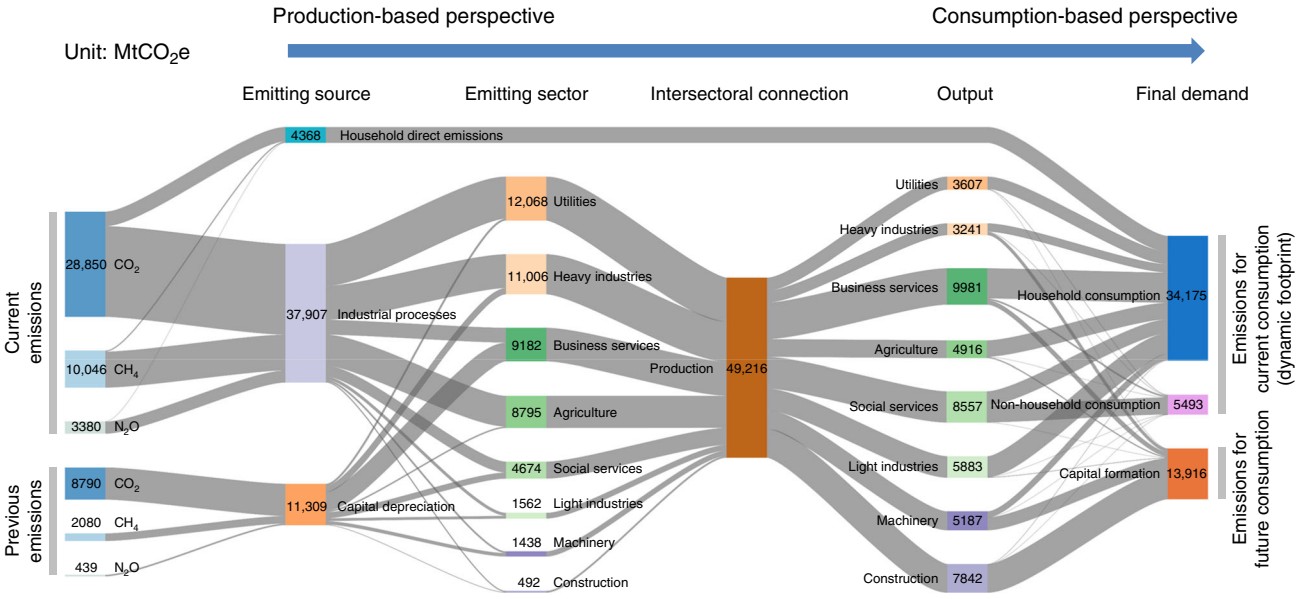

**Fig. 3** Sankey diagram of embodied GHG flows of the world in 2009 under dynamic footprint accounting. Capital stock is shown to be an important source as well as destination of GHG flows. The industrial structure from production-based perspective is very different from that from consumption-based perspective. Sector definition is reported in Supplementary Table 1

services sector constituted the largest fraction of emissions embodied in depreciated capital, which can be explained by the large capital investment and depreciation for real estate and transportations. The heavy industry sector and the social services

sector are the second and the third largest capital-related emissions sources owing to their considerable machinery and equipment usage. Regarding the composition of newly formed capital, over half the embodied emissions came from the

construction sector, which is obviously due to the durable feature of its outputs. The global dynamic footprint in 2009 is about one quarter smaller than the total emissions embodied in final demand of that year, while this gap was, through its inclusion in capital formation, embodied in future rather than current consumption.

**National emissions and footprints**. Per-capita dynamic footprints were lower than their traditional counterparts for most countries during 1995–2009, with only two exceptions of Japan and the UK (Fig. 4). The difference between PBA and traditional CBA comes from spatial emission transfers, i.e., a net exporter (importer) of embodied emissions has a traditional footprint that is lower (higher) than its territorial emissions. At the meanwhile, the difference between dynamic and traditional measures of CBA comes mainly from the temporal change in capital stock, i.e., the dynamic footprint is lower (higher) than the traditional footprint if there is a net increase (decrease) in emissions embodied in capital stock.

With an average discrepancy smaller than 5%, the traditional footprint was, in the long run, a close approximation of the dynamic indicator for some developed countries, e.g., France, Germany, Japan, the UK, and the US. However, this approximation does not hold for countries with rapid capital accumulation, since much higher levels of GHG emissions are embodied in capital formation than in capital depreciation, leading to a dynamic footprint that is significantly lower than the traditional one. In particular, for some fast-developing countries including China, India, Indonesia, Mexico, and Turkey, the discrepancy between the dynamic and traditional footprints was even larger than the difference between the traditional CBA footprint and the territorial PBA emissions. This suggests that the error created by neglecting capital inputs in the traditional model can be remarkable. Moreover, while traditional accounting suggests Ireland and Korea had higher consumption-based than production-based emissions as a result of their outflowing emission transfers, a proper accounting for changes in capital leads to the opposite finding. In business cycles, investment in capital usually falls during recessions and rises during economic

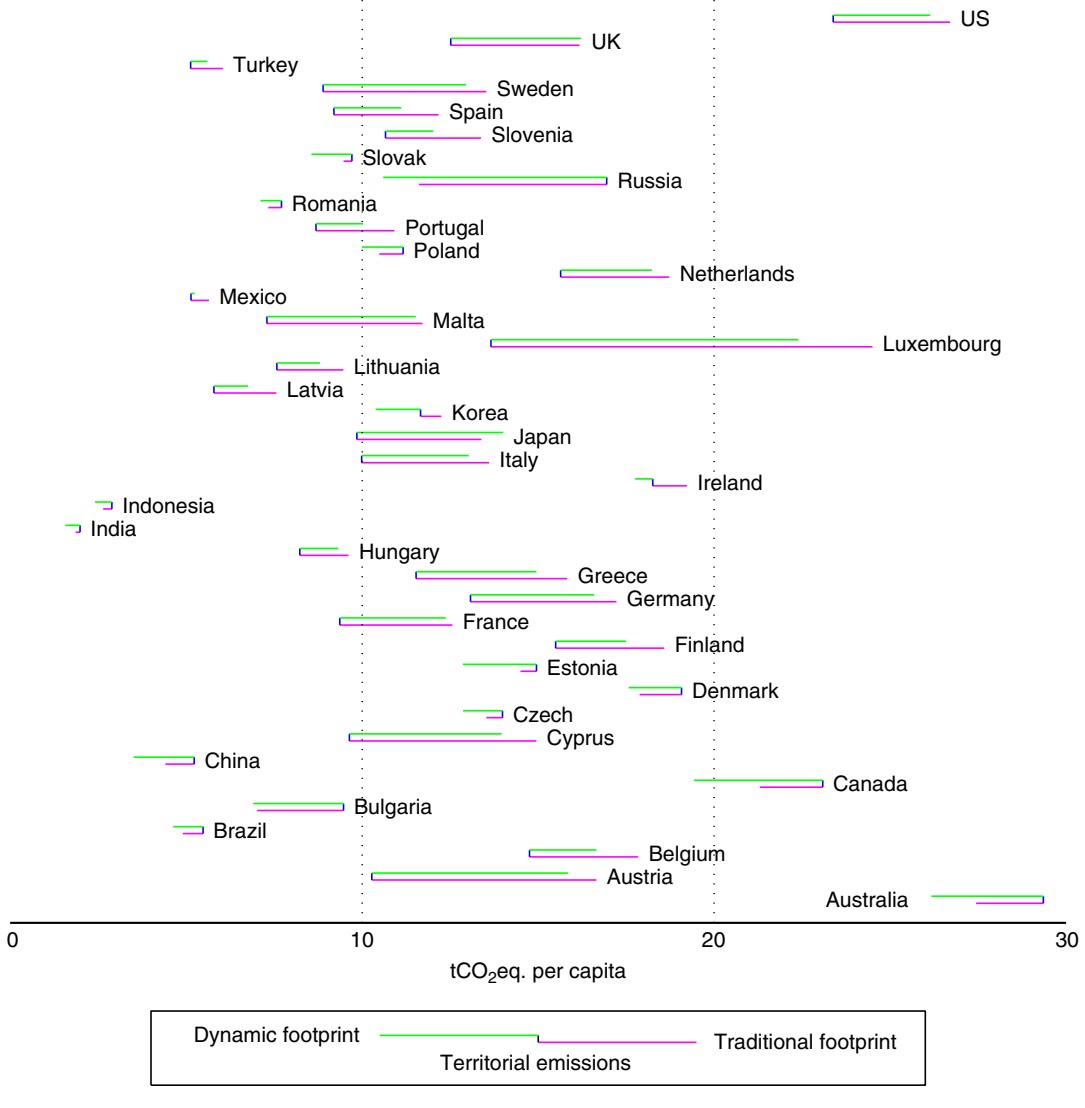

**Fig. 4** National per-capita territorial emissions, traditional footprint, and dynamic footprint (average of 1995–2009). The position of the vertical short line represents territorial PBA emissions; the free end of the upper line represents the country's dynamic footprint; and the free end of the lower line represents the traditional footprint. Values are indicated on the horizontal axis. By way of an example, China's dynamic footprint is lower than its traditional footprint, while both CBA measures are lower than is territorial PBA emissions

recoveries. Therefore, emissions embodied in newly formed capital are also more heavily affected by the economic cycle than those in depreciated capital (Supplementary Figure 1).

By acknowledging the contribution of capital through depreciation as well as investment, the dynamic footprint has a steadier temporal trend than the traditional footprint (Supplementary Figure 2). While traditional CBA attributes the gap between emission footprint and territorial emissions solely to spatial difference in production and consumption, the accounting method proposed in this study decomposes it into international trade and capital change (changes of capital stock and inventory). For China, as an example, net capital change contributed to over one third of the deviation between its dynamic footprint and territorial emissions during 1995–2009. In general, for economies with higher production-based than consumption-based emissions, large fraction of the gap can be explained by their capital accumulation. By contrast, capital change outweighs significant part of the net embodied emission import for economies whose consumption-based emissions are higher than their territorial counterparts (Supplementary Figure 3).

**Emissions embodied in capital stocks**. Along with the growth of the emissions embodied in global capital stock from 84 $GtCO_2e$ in 1995 to 116 $GtCO_2e$ in 2009, there was a general trend that saw the developing economies expanding their shares (Fig. 5). India had the fastest proportional growth in capital accumulation, while China had the largest increment in terms of emission embodiment during the period. In contrast, Japan had the largest reduction in capital. In spite of absolute increases in emissions embodied in capital stocks, the EU and the US both had smaller shares in 2009 compared with in 1995. The global financial crisis of 2007–2009 wrought severe shocks on the capital investments of these two economies, resulting in an accelerated decline in their

shares. Using the indicator decomposition technique[31], we find capital stock expansion was the dominant factor driving up the emissions embodied in capital stock during 1995–2009; international capital share change and emission intensity change both played increasingly important roles during the period (Supplementary Table 2).

Investment and depreciation continuously adjust the composition and emission intensity of capital stock in a dynamic way. The emission intensity of capital stock decreased slowly (average annual change rate of −1.0% for the 39 countries) during 1996-2009, but the trend varies (between −5.6% and +3.2%) across countries and years (Supplementary Figure 4 and Supplementary Table 4).

**Emissions embodied in trade**. Taking into account the emissions embodied in capital depreciation dramatically affects the composition and volume of emissions embodied in international trade, which in turn influence national trade balances in terms of embodied emissions (Supplementary Figure 5). Exactly how capital depreciation affects the emission intensity of exports varies greatly among countries. For example, in 2009, China had lower direct current emissions embodied in each dollar of export than did India. However, investment in capital (such as infrastructure, factories, and machinery) was more important in China than in India, leading to larger fraction of emissions embodied in associated capital depreciation. When the cumulative emissions associated with capital (which would have been accounted for in the year of emission under PBA approaches) are properly accounted for through depreciation in our dynamic CBA approach, China's higher overall emission intensity is again revealed. The results are consistent with the conclusion of a recent report[32], which indicated that capital investment can form a significant part of the total resources (and thus their associated

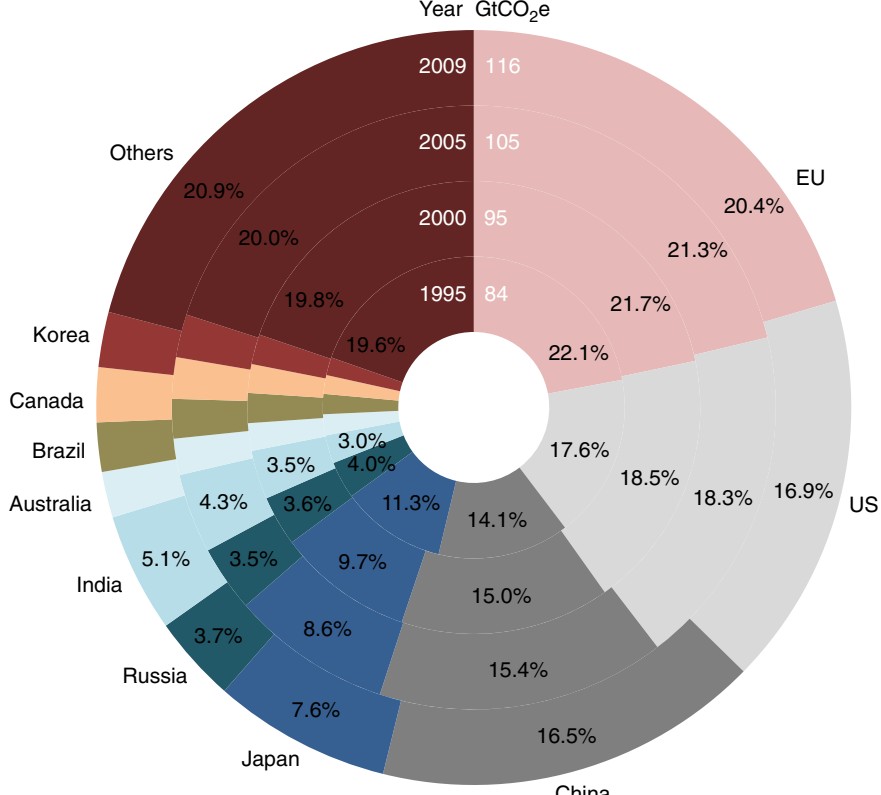

**Fig. 5** Global emissions embodied in capital stock in 1995, 2000, 2005, and 2009. Others includes Indonesia, Mexico, Turkey, and rest of the world, see Supplementary Table 3. Absolute value in $GtCO_2e$ shown represents global total

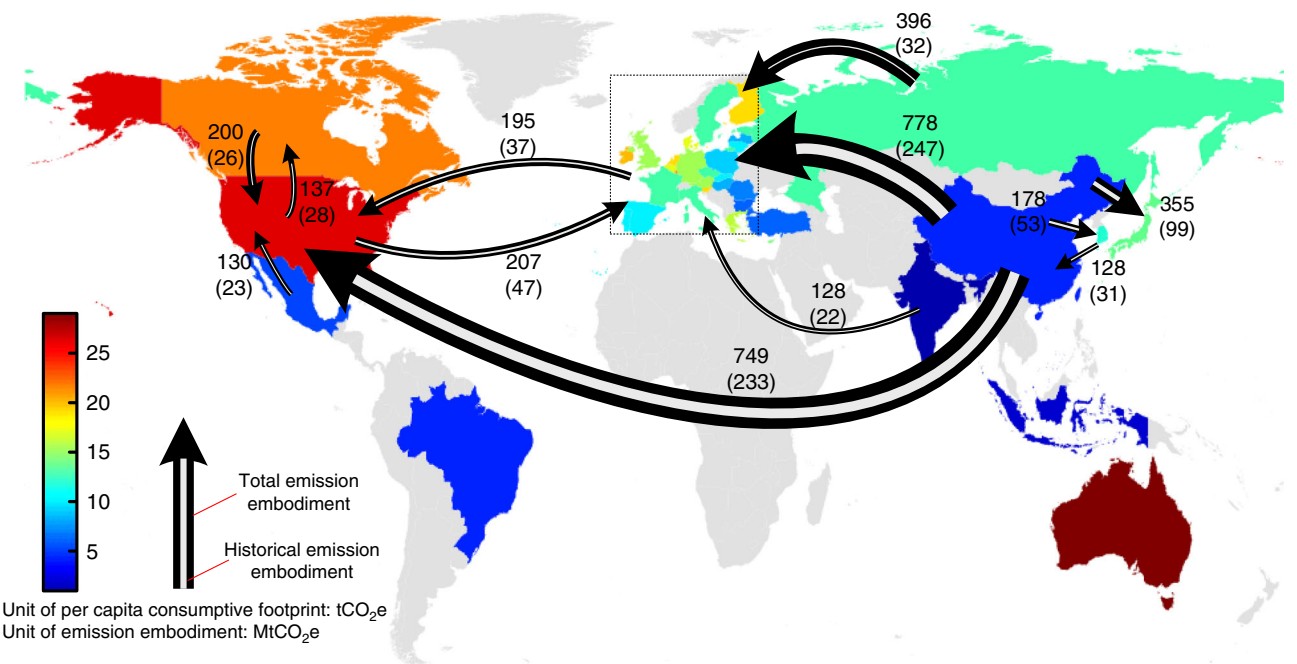

**Fig. 6** Per-capita dynamic footprint and emissions embodied in major bilateral trade flows in 2009. The region in the dotted box represents the EU. The color of the territory indicates national per-capita dynamic footprints. The numbers associated with bilateral trade are total embodied emissions (outside parentheses) and cumulative embodied emissions (inside parentheses). The arrow illustrates the GHG emissions embodied in the goods produced in one region and transported to another, but does not necessarily imply any net regional emissions change caused by the trade

emissions) attributable to international trade. In contrast to exports, the imports of different countries have less diversified emissions intensities.

As the world's largest exporter of embodied emissions, in 2009, China had over one-fifth of its export-related emissions flow to the EU and the US, respectively: the two largest importers of embodied emissions in the world (Supplementary Figure 5). The fact that regions with higher per-capita dynamic footprints usually had net embodied emissions inflows through their bilateral trade with regions with lower per-capita dynamic footprints (Fig. 6 and Supplementary Tables 5 and 6) reinforces the already large international disparity in terms of per-capita consumption-based emissions[11].

A dynamic CBA model is established by taking the changes of capital stock into account, which extends our understanding of the GHG emissions associated with consumption and investment within the CBA framework. In spite of the growing emissions embodied in international trade[17,33,34], the consideration of capital dynamics suggests that the gap between production-based and consumption-based emissions can be decomposed into international trade (the discrepancy between territorial emissions and traditional footprint) and capital accumulation (the discrepancy between traditional footprint and dynamic footprint). While the former redistributes emissions across geography, the latter redistributes emissions across time.

By employing the new model to investigate the global consumption-based GHG emissions during 1995–2009, our results show that the world's dynamic emissions footprint is 7.4% smaller than its territorial emissions. This means the assumption adopted in traditional CBA, i.e., consumption-based emissions are equal to production-based emissions at the global scale, might lead to systematic error for emission footprint estimation. However, it is essential to identify the very different roles of capital in shaping consumption-based emissions for regions with different development characteristics. In many developed economies, net capital stock change does not play a

remarkable role for the overall embodied GHG budget because a major part of the new capital investment is used to replace the depreciated capital. As a result, traditional footprint presents a close approximation of the dynamic footprint, which suggests both footprints can be adopted as fair indicators of the consumption-based emissions. On the contrary, in some developing countries, the new capital investment significantly outweighs the capital depreciation and thus leads to considerable capital stock change. Therefore, net capital accumulation embodies large volume of GHG emissions. In those countries, it is important to differentiate the emissions embodied in capital accumulation from consumption demand. This finding suggests we need to be very cautious for the interpretation and comparison of the traditional emission footprint results when developing countries are involved. In other words, it is a prioritized task to update the CBA for developing countries using the dynamic model.

## Methods

**Endogenizing capital in CBA**. Traditional CBA studies are implicitly assuming that the GHG emissions embodied in newly formed capital are equal to those embodied in depreciated capital, thereby maintaining a constant quantity of emissions embodied in capital. This assumption, however, ignores the dynamics of capital, which plays an important role in enabling and supporting production in industrialized economies[25,35–38]. Unlike the goods used for consumption, which are final outputs from the perspective of production chains[11], capital is an intermediate input for the economy. In order to depict patterns of embodied GHG emissions by incorporating the influence of capital stock change, the global multi-regional input-output system is closed in this study by endogenizing capital formation and depreciation from final demand and primary input. Lenzen and Treloar[39] presented two methods, i.e., the augmentation method and flow matrix method, for endogenizing capital change into a static input-output model. While the flow matrix method is theoretically more accurate, since it avoids the systematic errors of the augmentation method, its high data requirements have led most empirical studies to apply the other method[40,41]. Therefore, this study adopts the framework of the augmentation method to endogenize capital change. Further, the model applied in this study is dynamized to trace the change in emissions as well as emission intensity of the capital stock[42–49].

The distinction between the traditional and dynamic footprints originates from the different sources of emissions each one contains, which can be explained intuitively by comparing the sectoral balance equation of the two models. We can start with the traditional balance equation[50] of sector $i$:

$$g_i + \sum_j x_{j,i}\varepsilon_j = y_i\varepsilon_i \qquad (1)$$

where $i$ and $j$ are the global sector identities (e.g., if there are in total two regions and three sectors in each region, $i$ and $j$ are integrates between 1 and 6)[51], $g_i$ is the direct emissions of sector $i$, $x_{j,i}$ is the output from sector $j$ that is used by sector $i$, $\varepsilon_j$ and $\varepsilon_i$ are the emission intensities of outputs from sectors $j$ and $i$, and $y_i$ is the total output of sector $i$. The emission intensity of output from any sector can be obtained from the matrix expression:

$$\mathbf{E} = (\mathbf{Y} - \mathbf{X})^{-1}\mathbf{G} \qquad (2)$$

where $\mathbf{E}$ is the column vector of sectoral emission intensity, $\mathbf{Y}$ is the diagonalized vector of sectoral output, $\mathbf{X}$ is the transpose of multi-regional input-output transactions matrix, and $\mathbf{G}$ is the column vector of sectoral direct emissions.

Introducing the time identity of $t$ indicating the variable corresponding to period $t$, Eq. (1) can be generalized to include capital input (capital depreciation and inventory use) as:

$$g_{i,t} + \sum_j x_{j,i,t}\varepsilon_{j,t} + d_{i,t}\bar{\varepsilon}_{k,t-1} + c_{i,t}\varepsilon_{i,t-1} = \left(y_{i,t} + c_{i,t}\right)\varepsilon_{i,t} \qquad (3)$$

where $d_{i,t}$ is the capital depreciation by sector $i$ in period $t$, $\bar{\varepsilon}_{k,t-1}$ is the emission intensity of capital stock of region $k$ in period $t-1$, and $c_{i,t}$ is the total inventory use (sum of negative inventory change in absolute value) of output from sector $i$ in period $t$. Here we assume the capital depreciated in period $t$ has the same emission intensity as the capital stock of period $t-1$, while the inventory used in period $t$ has the same emission intensity as the output of period $t-1$. Notice that when the emissions embodied in capital input is ignored ($\bar{\varepsilon}_{k,t-1} = 0$) and inventory input has the same emission intensity as current output ($\varepsilon_{i,t} = \varepsilon_{i,t-1}$), Eq. (3) is regressed to Eq. (1).

The balance equation for capital stock of region $k$ can be formulated as:

$$s_{k,t-1}\bar{\varepsilon}_{k,t-1} - \sum_{l_k} d_{l_k,t}\bar{\varepsilon}_{k,t-1} + \sum_j f_{j,k,t}\varepsilon_{j,t} = s_{k,t}\bar{\varepsilon}_{k,t} \qquad (4)$$

where $l_k$ is the identity of sectors from region $k$, $s_{k,t-1}$ and $s_{k,t}$ are the capital stocks of region $k$ in periods $t-1$ and $t$, and $f_{i,k,t}$ is the output of sector $i$ that is used for capital formation by region $k$ in period $t$. When initial condition is required to determine the emission intensity of any concerned period based on Eqs. (3) and (4), we assume $\bar{\varepsilon}_{k,t_0+1} = \bar{\varepsilon}_{k,t_0}$ and $\varepsilon_{i,t_0+1} = \varepsilon_{i,t_0}$ for the first two periods (sensitivity analysis is provided in Supplementary Information regarding alternative initial condition assumptions). Therefore, Eq. (4) becomes:

$$\bar{\varepsilon}_{k,t_0} = \sum_j f_{j,k,t_0+1}\varepsilon_{j,t_0+1} \Big/ \left(s_{k,t_0+1} - s_{k,t_0} + \sum_l d_{l,t_0+1}\right) \qquad (5)$$

Note that the denominator of Eq. (5) is equal to capital formation of region $k$ ($\sum_i f_{i,k,t}$), then we have:

$$\bar{\varepsilon}_{t_0} = \sum_j f_{j,k,t_0+1}\varepsilon_{j,t_0+1} \Big/ \sum_j f_{j,k,t_0+1} \qquad (6)$$

Equation (6) implies that when emission intensity of capital stock is constant in two consequent periods, it is equal to the average emission intensity of capital formation in the latter period. Apply Eq. (6) and the initial condition assumptions to Eq. (3) for $t = t_0+1$ get:

$$g_{i,t_0+1} + \sum_j x_{j,i,t_0+1}\varepsilon_{j,t_0+1} + d_{i,t}\sum_j f_{j,k,t_0+1}\varepsilon_{j,t_0+1} \Big/ \sum_j f_{j,k,t_0+1}$$
$$= y_{i,t_0+1} \qquad (7)$$

It can be solved in matrix notation to get:

$$E_{t_0+1} = \left(Y_{t_0+1} - X_{t_0+1} - A_{t_0+1}\right)^{-1} G_{t_0+1} \qquad (8)$$

where $E_{t_0+1}$, $Y_{t_0+1}$, $X_{t_0+1}$, and $G_{t_0+1}$ are the topologies of $E$, $Y$, $X$, and $G$ for period $t_0+1$, and $A_{t_0+1} = [a_{i,j,t_0+1}]$. The components of $A_{t_0+1}$ can be obtained as:

$$a_{i,j,t_0+1} = \begin{cases} d_{i,t}f_{j,k,t_0+1} \Big/ \sum_m f_{m,k,t_0+1} & \left(\sum_m f_{m,k,t_0+1} \neq 0\right) \\ 0 & \left(\sum_m f_{m,k,t_0+1} = 0\right) \end{cases} \qquad (9)$$

where $m$ is also global sector identity. Notice Eq. (8) differs from Eq. (2) by taking the capital input ($A_{t_0+1}$) and time dimension into account, which leads to the different connotation of emission intensity in the traditional and dynamic accounting models. For $t = t_0+2$, solve Eq. (3) to get:

$$E_{t_0+2} = \left(Y_{t_0+2} - X_{t_0+2} + C_{t_0+2}\right)^{-1}$$
$$\left(G_{t_0+1} + D_{t_0+2}E_{t_0+1}^* + C_{t_0+2}E_{t_0+1}\right) \qquad (10)$$

where $C_{t_0+2}$ is the diagonalized vector of sectoral total inventory use in period $t_0+2$, $D_{t_0+2}$ is the diagonalized vector of sectoral depreciation in period $t_0+2$, and $E_{t_0+1}^* = [\bar{\varepsilon}_{i,t_0+1}^*]$ with $\bar{\varepsilon}_{i,t_0+1}^* = \bar{\varepsilon}_{k,t_0+1}$ whenever sector $i$ belongs to region $k$. Thereafter Eq. (4) can be solved for all regions for $t = t_0+2$ to get:

$$\bar{\varepsilon}_{k,t_0+2} = s_{k,t_0+2} \Big/ \left(s_{k,t_0+1}\bar{\varepsilon}_{k,t_0+1} - \sum_l d_{l,t}\bar{\varepsilon}_{k,t_0+1} + \sum_j f_{j,k,t_0+2}\varepsilon_{j,t_0+2}\right) \qquad (11)$$

For future periods, emission intensities of sectoral output and capital stock can be obtained by solving Eqs. (3) and (4) sequentially.

**Data source and manipulations**. The calculations in this study were executed using the World Input-Output Database[28] (which was chosen over the other multi-regional input-output databases because its Socio Economic Accounts contains detailed data on capital) and the World Bank[52]. The World Input-Output Database disaggregates the world into 41 (38 national, two subnational, and one multi-national) economies, with each economy divided into 35 sectors (detailed sector information can be found in Supplementary Table 5). The database provides sectoral capital stock data between 1995 and 2007 for 40 national and subnational economies; 24 of them have data extended until 2009 and another two until 2008 (detailed information can be found in Supplemental Table 3). While the perpetual inventory method[53] has been applied to connect capital stock with its depreciation and formation, the following manipulations have been conducted based on the original database.

First, the missing sectoral capital stock data are estimated by assuming that sectoral capital depreciation rate has been constant since 2007 (for those with original data until 2007) or 2008 (for those with original data until 2008). Second, the original sectoral capital stock data of Slovenia for 2008 and 2009 lead to extraordinary values (e.g., the capital stock drops by 22.7% from 2008 to 2009) in our calculation, which are deemed to be caused by unknown errors in data preparation. In response, the sectoral capital stock data of the two years are also estimated following the assumption of the first step. Third, the initial sectoral capital stock of the original multinational economy, i.e., the rest of the world, in 1995 is estimated by assuming it has the same sectoral capital stock to output ratio as the other 40 economies as a whole. Finally, the capital stock of the rest of the world between 1996-2009 is estimated by assuming this region has the same sectoral capital depreciation rate as the 40 economies as a whole.

After the above manipulations, there are still 219 (1.1% of the total) negative sectoral capital depreciation values left, which have been set to zero in our calculations. Furthermore, the multi-regional input-output table provided by the World Input-Output Database contains 22 (1.5%) null rows in the transaction matrix of each year, because corresponding sectors do not exist (physically or due to statistical difference) in the related economies. Since the null sectors will lead to infinite solutions to the dynamic as well as the traditional accounting model, the sectoral output of the 22 sectors will be set to an extraordinary large number ($1,000 trillion in our calculation), which can ensure a unique solution without introducing significant error.

One thing needs to bear in mind is, since the (dynamic) CBA model relies on large volume of data with high uncertainty, the calculated results need to be interpreted with cautious. For example, it is more reasonable to be used to identify the temporal change, the comparative importance of different sectors, or the directional discrepancy between models, but the footprint value is always too uncertain to be a meaningful indicator when it stands along.

**Other methodological issues**. It is worth noting that the description of our model as "dynamic" does not imply a predictive function, but emphasizes the perpetual accumulation and depreciation of capital. Therefore, the dynamic CBA model is used in this study as an ex-post approach to identify the role of capital in regional consumption-based GHG emissions accounting, instead of as an ex-ante technique to forecast future emissions.

While the broadest concept of capital includes a variety of components, data availability leads us in this study to adopt a narrower definition to cover only fixed capital and changes in inventories. Inventory is taken into account because of its capacity to redistribute emissions temporally (despite within a short period), which is similar to the effect of physical capital in dynamic footprint accounting. Two of the most important factors that have been overlooked are the investment in intellectual property, i.e., intellectual capital, and that in skills and education, i.e., human capital. For the former factor, the present study has potential to be

upgraded as more and more statistical authorities start to apply the latest 2008 SNA[54], which treats intellectual capital as a subcategory of fixed capital. For the latter factor, so far no generally accepted statistical solution has been advanced to preciously evaluate the stock and flow of human capital. However, it seems to be fair to assume the investment and stock change of human capital increase steadily in both developed and developing economies at the long run, thus the present results turn out to underestimate the GHG emissions embodied in global capital stocks.

Finally, the current study focuses on GHG emission patterns at the national and international levels by acknowledging the influence of capital stock change, while questions involving more direct consequential analysis, such as how specialized division of labor and production will affect international trade as well as emission transfers, are untouched. In order to move from traditional CBA towards more consequential accounting[55–57], Kander et al.[23] advanced a meaningful approach by a technology adjusted measure. The analysis of this study could also be conducted with similar extension, but we have chosen to take the traditional CBA as our starting point here to keep the analysis clean and accessible, and since separating the effects of investments from consumption is a novel approach that is interesting on its own.

## Data availability

The datasets generated and analyzed during the current study are available from the corresponding authors on reasonable request.

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

## Acknowledgements
We thank Michael Jakob for comments, discussions, and general support. This study has been supported by the Fundamental Research Funds for the Central Universities and the Research Funds of Renmin University of China (14XNJ011), National Science Fund for Distinguished Young Scholars of China (71725005), National Key Research & Development Program (2016YFA0602304), and National Natural Science Foundation of China (41501605 and 71874097).

## Author contributions
Z.-M.C. designed the research. Z.-M.C., S.O., M.J., B.C., D.G., S.X., and L.L. performed the analysis. M.L., T.W., B.C., Z.L., G.C., and X.Z. contributed to background. Z.-M.C., S.O., T.W., L.L., D.G., J.M., A.A., and T.H. drew the figures. All authors contributed to interpretation of results and writing.

## Additional information

**Competing interests:** The authors declare no competing interests.

