## [Peer Review File · Nature Communications]

Editorial Note: this manuscript has been previously reviewed at another journal that is not operating a transparent peer review scheme. This document only contains reviewer comments and rebuttal letters for versions considered at *Nature Communications*.

Reviewer #1

Remarks to the Author:

I find it difficult to agree with the authors that their dynamic CBA is more useful for policy than the traditional CBA. That is a more 'correct' measure of current-year consumption emissions is perhaps perfectly reasonable, given a specific definition. But in policy terms, traditional CBA gives us very clearly the current year's consumption emissions and all committed, future consumption emissions resulting from the current year's actual, production emissions. That is useful for policy. How is spreading current-year emissions over future years more useful for policy? By attempting to correct one of the many shortcomings of CBA, the new method relies on more data that is unreliable and uncertain, again diminishing its policy utility.

The authors have chosen to add a lengthy discussion of the policy relevance of CBA to the paper, largely building on my own previous review comments, apparently ignoring my statement "I do not believe it is necessary that this article present a discussion of these points in depth, but the current inconsistency must be cleared up." I do not think it is necessary or useful in this paper to make such a detailed defence of the policy relevance of CBA, and would recommend cutting this down substantially. The topic of the paper is most certainly not the policy relevance of CBA, and this reads as a significant and unwarranted digression.

On page 4 the authors discuss a 'major finding of this study', namely that global PBA is larger than global dynamic CBA in any given year. This seems to me something that would be relatively obvious as soon as you consider a dynamic CBA, and would be better discussed as a hypothesis that is confirmed by the analysis. As long as you have capital formation larger than depreciation of existing capital, this result should hold, given basic assumptions about emissions intensity of capital formation. I think the point would be much clearer if laid out in this way.

I despair at the frequent use of high-precision results, which indicate a lack of recognition of the uncertainty in the underlying data being used. The global dynamic footprint in 2009 should not be presented as 39,668 Mt.

The authors fail to provide even the most basic explanation for some of their results. Is it really sufficient to simply report that "the business services sector constituted the largest fraction of emissions embodied in depreciated capital" without any explanation for why this might be expected?

Detailed Responses to Reviewer' Comments

Authors:

Thank you very much for your insightful comments again! We have carefully revised the manuscript based on your comments in order to emphasize the methodological advancement and to avoid the lengthy as well as distracting policy discussion. Detailed changes and responses to your comments are listed as below. We hope the current version of our study can present a solid methodological contribution to the field of consumption-based greenhouse gas accounting.

Reviewer #1:

I find it difficult to agree with the authors that their dynamic CBA is more useful for policy than the traditional CBA. That is is a more 'correct' measure of current-year consumption emissions is perhaps perfectly reasonable, given a specific definition. But in policy terms, traditional CBA gives us very clearly the current year's consumption emissions and all committed, future consumption emissions resulting from the current year's actual, production emissions. That is useful for policy. How is spreading current-year emissions over future years more useful for policy? By attempting to correct one of the many shortcomings of CBA, the new method relies on more data that is unreliable and uncertain, again diminishing its policy utility.

Authors:

Thank you for pointing out the essential point that the core contribution of the new model is its "more 'correct' measure of current-year consumption emissions" instead of its policy utility. Accordingly, we have conducted major revision on the manuscript to emphasize its methodological development. Detailed changes include:

- (1) We revised the title to "Consumption-based Greenhouse Gas Accounting with Capital Stock Change" to make the methodological advancement more clear.
- (2) We completely rewrote Abstract to emphasize improvements of the new model over the traditional one based on both theoretical and empirical perspectives. For example, the second sentence ("While traditional accounting... and temporal capital accumulation.") points out the theoretical improvement by the consideration of capital dynamics. The last

four sentences (“Results show... for fast-developing countries.”) explain the different empirical results obtained by the new and traditional models and identify that fast-developing countries are most affected by the model switching.

- (3) In the last sentence of Introduction, the purpose of this study is concluded clearly as “to bring forward a new method to describe the connection between consumption and GHG emissions by acknowledging the influence of capital stock changes in the developing real-world scenario.”
- (4) We also completely rewrote Discussion as an extension and echo to Abstract. In the first paragraph (“A dynamic CBA model... redistributes emissions across time.”), the theoretical difference between the new model and the traditional one is discussed. In the second paragraph (“By employing the new model... using the dynamic model.”), the different empirical results are illustrated in more details.

Reviewer #1:

The authors have chosen to add a lengthy discussion of the policy relevance of CBA to the paper, largely building on my own previous review comments, apparently ignoring my statement "I do not believe it is necessary that this article present a discussion of these points in depth, but the current inconsistency must be cleared up." I do not think it is necessary or useful in this paper to make such a detailed defence of the policy relevance of CBA, and would recommend cutting this down substantially. The topic of the paper is most certainly not the policy relevance of CBA, and this reads as a significant and unwarranted digression.

Authors:

Thank you for your suggestions, according to which we have tailored the text and cut down all the detailed defence of the policy relevance. More specifically, we have removed the following discussion from the previous manuscript. (1) Policy implications at the concluding paragraph of Introduction. (2) The defence of general policy usefulness of consumption-based accounting from the second to the fifth paragraph of Discussion. (3) The policy utility of the new model at the last three paragraphs of Discussion.

We are very sorry for not completely following your previous comments “I do not believe it is necessary that this article present a discussion of these points in depth, but the current inconsistency must be cleared up.” In the previous revision, we only focused on the clearing up suggestion, i.e., removing the inconsistent statements about consumption-based accounting. But we added

lengthy discussion to defend the policy relevance with an attempt to satisfy previous Referee 2. Now we got to notice that it is an inappropriate strategy of trying to satisfy (some of) the referees but ignoring the more essential audiences, i.e., readers do not need the lengthy policy discussion in this methodological piece. We feel sorry about that and hope the new revision is moving towards a correct direction.

Reviewer #1:

On page 4 the authors discuss a 'major finding of this study', namely that global PBA is larger than global dynamic CBA in any given year. This seems to me something that would be relatively obvious as soon as you consider a dynamic CBA, and would be better discussed as a hypothesis that is confirmed by the analysis. As long as you have capital formation larger than depreciation of existing capital, this result should hold, given basic assumptions about emissions intensity of capital formation. I think the point would be much clearer if laid out in this way.

Authors:

Thank you for the suggestion! Accordingly, we have revised the laid out to make the point much clearer:

“Given the basic assumption that the emissions intensity of capital stock is not changing dramatically over a short period (which will be shown to be true later), one would expect global PBA is larger than global dynamic CBA footprint when the capital formation exceeds the depreciation of existing capital. This hypothesis is confirmed by the empirical results regardless of the type of GHG or the years studied (Figure 2).” (Lines 81-85, Page 4 in the revised manuscript)

Reviewer #1:

I despair at the frequent use of high-precision results, which indicate a lack of recognition of the uncertainty in the underlying data being used. The global dynamic footprint in 2009 should not be presented as 39,668 Mt.

Authors:

Thank you for the suggestion! We rewrote several parts of the manuscript (including but not limited to Abstract and the paragraph about the global dynamic footprint) to avoid the high-precision result presentation. Despite

quantitative analyses are still presented in some occasions, they are kept to a minimum. Besides, one paragraph is added to remind readers about the interpretation of results with consideration of the high uncertainty along with them. Some of the related changes are now provided as:

“Here we develop a dynamic model to incorporate capital stock change in consumption-based greenhouse gas (GHG) emissions accounting. While traditional accounting attributed the gap between consumption-based and production-based emissions solely to international trade, the dynamic model decomposes it into both spatial emission transfer and temporal capital accumulation. The new model is applied using the global data for 1995-2009. Results show the global emissions embodied in consumption calculated by the new model are smaller than that by the traditional model. The emissions embodied in global capital stock increased steadily during the period. However, capital plays very different roles in shaping consumption-based emissions for economies with different development characteristics. As a result, the dynamic model presents similar consumption-based emissions estimation for many developed countries comparing with the traditional model, but it significantly improves the results of the latter for fast-developing countries.” (Lines 8-19, Page 1 in the revised manuscript)

“Under the dynamic CBA analysis, in 2009, the GHG emissions embodied in capital formation is significantly larger than that embodied in depreciation (Figure 3). This highlights the importance and influence of capital in global production chains. Utilities and heavy industry released almost half of the global territorial GHG emissions in 2009, but only a very small fraction of the world’s dynamic footprint came from these two sectors. This is because a majority of their outputs was used as intermediate inputs, not as consumption goods. The emissions emanating from the business services sector constituted the largest fraction of emissions embodied in depreciated capital, which can be explained by the large capital investment and depreciation for real estate and transportations. The heavy industry sector and the social services sector are the second and the third largest capital-related emissions sources owing to their considerable machinery and equipment usage. Regarding the composition of newly formed capital, over half the embodied emissions came from the construction sector, which is obviously due to the durable feature of its outputs. The global dynamic footprint in 2009 is about one quarter smaller than the total emissions embodied in final demand of that year, while this gap was, through its inclusion in capital formation, embodied in future rather than current consumption.” (Lines 105-121, Page 5 in the revised manuscript)

“One thing needs to bear in mind is, since the (dynamic) CBA model relies on large volume of data with high uncertainty, the calculated results need

to be interpreted with cautious. For example, it is more reasonable to be used to identify the temporal change, the comparative importance of different sectors, or the directional discrepancy between models, but the footprint value is always too uncertain to be a meaningful indicator when it stands along.” (Lines 378-383, Page 13 in the revised manuscript)

Reviewer #1:

The authors fail to provide even the most basic explanation for some of their results. Is it really sufficient to simply report that "the business services sector constituted the largest fraction of emissions embodied in depreciated capital" without any explanation for why this might be expected?

Authors' responses:

Thank you for the comments! We have tried to explain the results carefully. For example, to explain why some sectors have large emissions embodied in depreciation while some other contributes to emissions embodied in capital formation, we revised the manuscript as:

“The emissions emanating from the business services sector constituted the largest fraction of emissions embodied in depreciated capital, which can be explained by the large capital investment and depreciation for real estate and transportations. The heavy industry sector and the social services sector are the second and the third largest capital related emissions sources owing to their considerable machinery and equipment usage. Regarding the composition of newly formed capital, over half the embodied emissions came from the construction sector, which is obviously due to the durable feature of its outputs.” (Lines 111-118, Page 5 in the revised manuscript)

REVIEWERS' COMMENTS:

Reviewer #1

I'm glad to see the improvements made to the paper in this last round.

I have only one remaining comment, and that is with regard to the second paragraph in the main text (lines 39-54 of page 3). Here is outlined the rationale for the dynamic CBA, but I do not feel it is sufficiently clear, and I confess from reading this I do not have a clear understanding of why this is important.

As it stands, the text states what conventional CBA does, says that this 'presents a very good approximation of reality' when economies evolve slowly, but that when economies evolve more quickly this can result in 'bias'.

But this discussion lacks an explanation of why it is wrong to assign 'consumption emissions' to the generation of capital, that is, in the year that capital is formed. It assumes that it is correct to assign them instead to the use of capital (via depreciation), if I've understood correctly. But why is one more correct than the other? This perhaps comes down to a chosen definition of what 'consumption emissions' actually means, and that must be spelled out here, because consumption emissions already has an accepted definition in the literature. If you're going to redefine it, then you should do that explicitly, rather than say that the previous measurement was wrong.

I understand that in a slowly evolving economy (one might say near steady-state), the formation of capital balances use/depreciation of capital in each year. Then under the authors' definition of 'correct' consumption emissions, the assumption of conventional CBA provides unbiased estimates.

So it seems clear what the difference is, but the authors fail to clearly argue why one is actually wrong and the other right.

If one defines consumption of capital as depreciation of capital (which is only the financial consumption of capital), and then claims that consumption emissions should be assigned to the place, sector, and time of consumption, then the authors' argument holds, but they must spell this out.

Detailed Responses to Reviewer's Comments

Reviewer #1:

I'm glad to see the improvements made to the paper in this last round.

I have only one remaining comment, and that is with regard to the second paragraph in the main text (lines 39-54 of page 3). Here is outlined the rationale for the dynamic CBA, but I do not feel it is sufficiently clear, and I confess from reading this I do not have a clear understanding of why this is important.

As it stands, the text states what conventional CBA does, says that this 'presents a very good approximation of reality' when economies evolve slowly, but that when economies evolve more quickly this can result in 'bias'.

But this discussion lacks an explanation of why it is wrong to assign 'consumption emissions' to the generation of capital, that is, in the year that capital is formed. It assumes that it is correct to assign them instead to the use of capital (via depreciation), if I've understood correctly. But why is one more correct than the other? This perhaps comes down to a chosen definition of what 'consumption emissions' actually means, and that must be spelled out here, because consumption emissions already has an accepted definition in the literature. If you're going to redefine it, then you should do that explicitly, rather than say that the previous measurement was wrong.

I understand that in a slowly evolving economy (one might say near steady-state), the formation of capital balances use/depreciation of capital in each year. Then under the authors' definition of 'correct' consumption emissions, the assumption of conventional CBA provides unbiased estimates.

So it seems clear what the difference is, but the authors fail to clearly argue why one is actually wrong and the other right.

If one defines consumption of capital as depreciation of capital (which is only the financial consumption of capital), and then claims that consumption emissions should be assigned to the place, sector, and time of consumption, then the authors' argument holds, but they must spell this out.

Authors:

Thank you for your additional comments. We have rewrote the introduction part to clarify the rationale of our new model.

In the revised manuscript, the reason why the new model provides better estimation than traditional one is justified based on the following logic. First, consumption-based emissions is defined as the environmental impacts generated during the production

and use of its final goods and services. Then, the dominating indicator, i.e., emissions embodied in final demand, applied in traditional CBA studies is introduced. The advantage (simplification of CBA calculation) and disadvantage (possible bias when certain conditions are not satisfied) of the indicator are briefly discussed. Afterward, we explain why the new dynamic model is an improvement of the traditional one to address the calculation bias problem. Finally, the modelling exercise (application of the dynamic model) and the key finding (improvement of the dynamic model is confirmed) of this study are introduced. Related changes are provided in the revised manuscript as:

“CBA studies ground on the opinion that those who benefit from a process should bear (at least some) responsibility for the associated emissions¹¹. Accordingly, one of the main tasks of CBA is to estimate the consumption-based emissions of an economy, i.e., the environmental impacts generated during the production and use of its final goods and services^{12,13}...”

...Within the multi-regional input-output framework, economic outputs are categorized as either intermediate input or final demand (which includes consumption and capital investment). In most traditional CBA studies, global GHG emissions of a given year are allocated to different countries according to their final demands (both consumption and capital investment) and thus current emissions embodied in final demand (usually termed emission footprint) of a country are used to indicate its consumption-based emissions^{11,25-29}. By adopting this allocation scheme and the associated indicator, CBA model is significantly simplified, because intertemporal dynamics through the use of past capital to enable current production are ignored and all domestic final demand is attributed to support domestic consumption.

The indicator of emissions embodied in final demand applied in traditional CBA model presents a good reflection of emissions driven by consumption activities for a country if the economy is near a steady state and if its capital is only used to produce domestically consumed goods. Unfortunately, in the real world, most economies are moving away from instead of towards the two conditions. Therefore, a traditional CBA perspective might bring bias to the estimation of consumption-based emissions because the capital dynamics and the international engagement of capital services are overlooked. As a result, this study revisits the definition of consumption-based emissions and constructs an alternative indicator by acknowledging the role and dynamics of capital stock in the global production structure. A new dynamic model is proposed by endogenizing capital as a dynamic stock within the multi-regional input-output framework, in which the contributions of capital input from different times as well as from different regions are taking into account to measure the emissions generated by consumption activities.

The new model is applied to the global economy based on the World Input-Output Database³⁰ to examine the role of capital in consumption-based GHG emissions

accounting. Our results confirm the evolution of global capital stock and the international engagement of capital use have significant impact to the consumption-based emissions, especially for fast-developing countries.” (Lines 27-64, Page 2 in the revised manuscript)